# Developing a Scientific Literacy Assessment Instrument for Portuguese 3rd Cycle Students

**Marcelo Coppi \*** , **Isabel Fialho** and **Marília Cid**

Center for Research in Education and Psychology, University of Évora, 7004-516 Évora, Portugal;
ifialho@uevora.pt (I.F.); mcid@uevora.pt (M.C.)
\* Correspondence: mcoppi@uevora.pt

**Abstract:** Teachers and researchers, in accordance with the main Portuguese curriculum documents in the area of Physical and Natural Sciences, agree that the development of scientific literacy is an integral part of basic education and report that the teaching of these skills is taking place in schools. However, few scientific literacy assessment instruments are available to assess students' proficiency in using these skills. In this article, we describe the design and processes for gathering validity evidence for the development of the Avaliação da Literacia Científica Essencial (ALCE) instrument. The ALCE assesses scientific literacy skills of students at the end of the 3rd cycle of Basic Education, in the cognitive domains of understanding; analysing and evaluating phenomena; problems and everyday situations involving content knowledge and skills developed in the subjects of Natural Sciences and Physical Chemistry. Our validity argument, which includes the gathering of evidence based on the content and internal structure of the instrument and is grounded in the current literature on the validation of assessment instruments, supports the use of the instrument for the assessment of students' scientific literacy level at the end of the 3rd cycle of Basic Education. The ALCE may be a useful tool to identify possible gaps between the teaching objectives and the students' scientific literacy proficiency and reflect on the methodologies, lesson plans and strategies used in the classroom in order to change them to better develop the students' scientific literacy.

**Keywords:** ALCE; basic education; assessment instrument; learning objectives; validity

## 1. Introduction

Scientists, science subject teachers and policy makers recognise that developing students' scientific literacy is one of the main goals of science education. However, in the relevant literature, there is no universally accepted definition of scientific literacy among experts in the field, but several characterisations include skills in scientific enquiry, content knowledge and attitudes towards science [1,2].

The Organisation for Economic Co-operation and Development (OECD), for example, defines scientific literacy as "the ability to engage with science-related issues, and with the ideas of science, as a reflective citizen. A scientifically literate person is willing to engage in reasoned discourse about science and technology" [3]. For Miller [4], one of the first researchers to propose a definition for the term, scientific literacy encompasses mastering basic scientific concepts and understanding the scientific enterprise and the impact of science and technology on society.

Bybee [5] interprets scientific literacy as "an individual's scientific knowledge and use of that knowledge to identify scientific questions, to explain scientific phenomena, and to draw evidence-based conclusions about science-related issues" (p. 65). DeBoer [1], on the other hand, proposes that we should accept the fact that scientific literacy is a broad concept that comprises several historically significant educational aspects that have changed over time and that it should therefore be perceived as synonymous with the public understanding of science.

Although it is presented as a polysemic term, all definitions of scientific literacy highlight students' competencies in understanding scientific processes and using science knowledge in real everyday situations [1,6–8]. It highlights the fact that understanding and knowledge should be broad and functional for general education rather than preparation for specific scientific and technical careers [1]. Furthermore, the concept of scientific literacy should contain knowledge and skills pertinent to every stage of the life cycle of scientific information, including an understanding of the scientific enterprise and the relationship of the products of science to society, and an understanding of science dissemination processes and their assimilation by the public [9].

In this regard, science educators, in general, express the indispensability of developing new ways of teaching for the development of scientific literacy. These should address the complexity of the consonance between science, as well as its products, and everyday life and apply useful pedagogical practices that empower students for the scientific and technological challenges of society [10–12].

The acquisition of scientific literacy skills has been the focal point of science education in several countries [1,13–15]. Coinciding with this core is the interest in discovering mechanisms and instruments to assess the development of students' scientific literacy skills [16,17].

Up to now, several instruments for assessing scientific literacy have been developed. However, many of them assess individual aspects of scientific literacy and subject-specific competencies and are targeted at secondary and university students [16–19]. It is also evident that a large number of studies are devoid, partially or totally, of the presentation of the validation processes of the assessment instruments [17–19], implying limitations for the use of the research results.

Regarding the 3rd cycle of Basic Education, there are few scientific literacy assessment instruments in the literature developed for this audience or equivalent [16,18]. In the literature review conducted by Coppi et al. [18], the authors showed that, of the 13 instruments that were identified, only 3 were intended for students in this education cycle, hence pointing out the need to develop new instruments for assessing scientific literacy for this school cycle.

In Portugal, the 3rd cycle of Basic Education presents its own particularities that justify the interest for the development of the students' scientific literacy. This, besides providing the acquisition of fundamental knowledge for the continuation of studies in Secondary Education, is marked, for example, by changes in the methodologies and strategies used in class, by the specialisation of content, by the number of scientific subjects and, consequently, by the number of teachers lecturing them.

In the transition from the 2nd cycle to the 3rd cycle, the students stop attending only one subject with specificity for scientific literacy (Natural Sciences), taught by a unique teacher, and start attending two subjects (Natural Sciences and Physical Chemistry), with their own characteristics and specific teachers. Furthermore, the 3rd cycle represents the last cycle in which it is compulsory for students to attend scientific subjects. In Secondary Education, only the students who opt for the Science and Technology course will attend classes of subjects aimed at the development of scientific literacy (Biology and Geology as well as Physics and Chemistry).

It was within the framework of these assumptions that the study presented in this article aimed to describe the development process of the Avaliação da Literacia Científica Essencial (ALCE), as well as the results of its application to students of the 3rd cycle of Portuguese Basic Education. Therefore, we designed the following research questions: (1) How can validity evidence for a scientific literacy assessment tool for students at the end of the 3rd cycle of Basic Education be developed and gathered? (2) Is ALCE able to gather enough validity evidence to enable its results to be used for decision making? (3) What is the scientific literacy level of students at the end of the 3rd cycle of Basic Education? (4) How many students are scientifically literate? (5) How might ALCE be used by teachers?

The ALCE Is an instrument consisting of 34 items in the *true-false-don't know* format, which are contextualized around real-world situations and problems, such as appreciating the importance of fossil records found in rocks for building the geological memory of planet Earth. Briefly, the process of development of the ALCE included the articulation of the scientific literacy skills of the main Portuguese curriculum documents in force until then and a benchmark of scientific literacy skills; the gathering of validity evidence based on content and internal structure of the instrument, through the application of the pilot test; the refinement of the instrument, taking into account the results of the expert and psychometric review processes; and the application of the final version of the instrument to students from several schools in mainland Portugal.

When developing the ALCE, we considered the four scientific domains present in the curriculum of the 3rd cycle of Basic Education (Biology, Geology, Physics and Chemistry). Whenever possible, we tried to assess them in such a way that the students' answers did not depend on exclusive knowledge of the content of the subjects (for example: What is photosynthesis? What is the speed of light? How many years ago did life appear on Earth? What is the atomic number of the chlorine atom?) but instead used the scientific skills developed to solve problems and explain everyday situations.

This decision is based on the definition of scientific literacy adopted by the authors, which refers to the development of scientific-technological competencies necessary for understanding science, explaining natural phenomena and solving daily problems; for actively participating in debates related to scientific issues; and for understanding the use and impacts of their application in society. In this sense, the development of the instrument aimed not only to assess the knowledge of scientific contents but also the use of these contents in real and everyday situations.

Our framework for conceptualising scientific literacy refers back to three dimensions of scientific literacy proposed by Miller [4] (nature of science, content of science and impact of science and technology on society), whose study was a precursor to most research conducted in order to conceptualise and assess it [19].

The dimension of the nature of science (NOS) encompasses the understanding of the scientific enterprise, attending to the assimilation of the processes, phases and products of scientific research [4]. It is assumed that the methods and techniques used in observation, reasoning, experimentation, interpretation, validation and dissemination of the results of a scientific study, in addition to characterising it as scientific knowledge, differentiate it from other types of knowledge [19,20].

Over the last decades, the nature of science has been widely adopted as a central element for the development of scientific literacy [21–23], characterized as a critical component for its development [24]. For Khishfe [24], the teaching of NOS skills can be carried out in three different contexts: history of science, scientific inquiry, and socioscientific issues. McComas and Clough [22], on their part, state that the nature of science "addresses issues such as what is science, how science works (including issues of epistemology and ontology), how science impacts and is impacted by society, and what scientists are like in their professional and personal lives" (p. 5).

However, Höttecke and Allchin [21] foresee two challenges in teaching competencies related to the nature of science: the teaching of NOS in the classroom is usually reduced to a restricted list of descriptive principles about science; most current approaches to the nature of science focus strictly on issues internal to science only, disregarding external factors such as funding, public understanding of science, and the impacts of scientific knowledge on the economy, ethics, environmental sustainability, politics, and other aspects of culture. These facts make the inclusion of items related to the nature of science in ALCE even more important.

The content of science dimension (CS), on the other hand, comprises the acquisition and understanding of an elementary scientific vocabulary, basic scientific terms and expressions and minimum scientific content [4]. These competencies are fundamental for dealing with everyday situations and problems and for participating in public discussion of issues

related to science and technology. Miller [4] states, for example, that "the individual who does not comprehend basic terms like atom, molecule, cell, gravity, or radiation will find it nearly impossible to follow the public discussion of scientific results or public policy issues pertaining to science and technology" (pp. 38–39).

Finally, the dimension of the impact of science and technology on society (ISTS) is related to the understanding of the increasing presence and influence of these two areas in everyday life and in public policy discussions. Miller [4] emphasizes the discernment of the positive and negative impacts of the use of the products of scientific-technological knowledge as well as the awareness of how public policies may compromise and affect the conduct of the employment of science and technology in society. The author stated that in the United States, already in the 1980s, science became

> [...] increasingly dependent upon public support, and as public regulation reaches deeper into the conduct of organized science, the frequency and importance of science policy issues on the national agenda will undoubtedly increase. Slightly over half the bills introduced in Congress involve science or technology in some degree, 25 and the establishment of the standing Committee on Science and Technology in the House of Representatives attests to the importance of scientific and technological issues in the national political system.　　　(p. 40)

It is important to note that the ALCE was not developed to be used as a large-scale assessment instrument but as a tool to collaborate with teachers' work. In this sense, we restricted our framing of the ALCE to these three dimensions to what we felt could be assessed in 3rd cycle Basic Education students, in order to provide information capable of assisting teachers in their decision making regarding their lesson plans, methodology and assessment.

We have omitted other dimensions from our framework, such as motivation and belief [16], not because we undervalue their components, but because we believe that they would require an instrument with a different type of configuration, items and benchmarks. Therefore, we acknowledge that our conceptualisation framework of scientific literacy is limited to the dimensions that we believe could be adequately assessed using the instrument we developed.

## 2. Instrument Development (ALCE)

### 2.1. Process Overview: Validity

Obtaining the validity of the ALCE was an important part of the instrument development process. Validity is arguably the most important property for ensuring assessment quality [25–29]. Russel and Airasian [29], for example, state that "the single most important characteristic of good assessment is its ability to help the teacher make appropriate decisions. This characteristic is called validity" (p. 26).

Traditionally, the concept of validity corresponds to the capacity of an instrument to measure what it was designed to measure, through three types of validity: content, construct and criterion validity [28,30–32]. However, our approach to developing the ALCE was grounded in a more recent unitary concept in the literature, proposed in the Standards for Educational and Psychological Testing [33], in which validity corresponds to "the degree to which evidence and theory support the interpretations of test scores for proposed uses of tests" (p. 11).

From this perspective, the types of validity that existed up to then are now accepted as five types of validity evidence, capable of supporting the argument for the use of the results of an assessment instrument, namely evidence based on test content; evidence based on response processes; evidence based on internal structure; evidence based on relations to other variables and evidence based on consequences of tests [32,33].

Therefore, we state that, for the development of the ALCE, we used several means to determine the validity of the instrument, focusing on the gathering of evidence based on the content and internal structure of the instrument. Validity evidence based on content may include logical or empirical analyses of the appropriateness of the instrument in relation to

the domain of the content addressed, the relevance of this domain for the interpretation of the assessment results, and also expert analysis regarding the correspondence between the items and the content addressed, the wording of the statements and the degree to which the items are demanding, for example [28,33]. Validity evidence based on internal structure, on the other hand, results from the statistical analysis of items and scores in order to investigate the primary and, if any, secondary dimensions assessed using an assessment instrument [34,35].

### 2.2. Gathering Validity Evidence Based on Test Content

Throughout the process of gathering validity evidence based on test content for the development of the ALCE [36], we drew on the stages proposed by Pasquali [37], presented in Table 1.

**Table 1.** Stages in the ALCE validity evidence based on test content gathering process.

| Stages | Designation |
|---|---|
| 1 | Definition of the cognitive domains |
| 2 | Definition of the scope of the content |
| 3 | Definition of the representativeness of the content |
| 4 | Drawing up the specification table |
| 5 | Construction of the instrument |
| 6 | Theoretical analysis of the items |
| 7 | Empirical analysis of the items |

Source: elaborated by the authors.

#### 2.2.1. Definition of the Cognitive Domains

We began the first stage by identifying and defining the cognitive domains that it was intended to assess. Cognitive domains or processes are defined as strategies through which knowledge is acquired, assimilated or used to solve problems and, thus, we used the following domains presented in Bloom's updated Taxonomy [38]: understand, which implies constructing meaning from instructional messages; analyse, which presupposes breaking down the content and determining the relationship of each part to the perception of the whole; and evaluate, which expresses the ability to establish a value judgement and make decisions based on criteria and standards.

The determination of the cognitive domains that make up the ALCE was based on the definition of scientific literacy that was established on the basis of the concepts previously discussed, which corresponds to the

> [...] understanding of the scientific enterprise and the conscious use of scientific and technological knowledge to solve problems, explain natural phenomena of everyday life and to participate actively in debates on scientific issues involving society, enabling the individual to act as a citizen. ([36], p. 104)

#### 2.2.2. Definition of the Scope of the Content

For the second stage of the process of gathering validity evidence, we delimited the content to be assessed using the ALCE, given that the items of the instrument are capable of reproducing only a representative sample of the content [35]. Considering the Portuguese context, we selected the four main curriculum documents of the area of Physical and Natural Sciences of the 3rd cycle of Basic Education, then in force: Curricular Guidelines for the 3rd cycle of Basic Education—Physical and Natural Sciences [39]; Essential Apprenticeships of Natural Sciences [40–42]; Essential Apprenticeships of Physical-Chemistry [43–45]; and Profile of Students Leaving Compulsory Schooling (PASEO) [46].

The option of including the Curricular Guidelines for the 3rd cycle of Basic Education in the universe of the content to be assessed is justified by the fact that this document specifies scientific literacy as the main objective of science teaching in the 3rd cycle of Basic Education and defines the competencies that should be developed by students by the end

of this cycle [39]. Using the contents of this document allowed us to assess the scientific literacy skills established by the Portuguese Ministry of Education.

In the case of the Essential Learnings of Natural Sciences and of Physical Chemistry, we reasoned our choice on the fact that these are guiding documents for both subjects, corresponding to a set of fundamental knowledge, skills and attitudes for students [40–45]. Supported by the Essential Learnings, the ALCE provides the assessment of substantial and essential content for the development of scientific literacy.

The decision to integrate PASEO in the scope of ALCE's contents was sustained by the fact that it is a "reference document for the organisation of the whole education system, contributing to the convergence and articulation of decisions inherent to the various dimensions of curriculum development" ([46], p. 8). Although it does not consist of contents per se, PASEO determines areas of competencies, understood as the confluence of knowledge, skills and attitudes, for the development of different literacies, including scientific literacy [46]. Thus, relying on this curriculum document made it possible to establish a concordance between the chosen cognitive domains and the essential skills to be developed in students throughout compulsory education.

In addition to these four curriculum documents chosen for the definition of the scope of the content, we considered it necessary to delimit the content based on scientific literacy guidelines already existing in the literature, since these encompass a vast amount of content and skills. We therefore chose to select the Benchmarks for Science Literacy (BFSL) [47], which correspond to the second product of Project 2061, from Science For All Americans (SFAA) [20], responsible for reformulating previously published science literacy objectives. This choice was based on the fact that these guidelines are supported by the dimensions of scientific literacy proposed in Miller's [4] study, which has provided the basis for several studies in the United States, the United Kingdom, the European Community, China, Canada and Japan [19] and also for the development of the instrument presented in this study.

Having identified the curriculum documents, we performed a documental analysis, in three phases, in order to select the content. The first phase consisted of listing all the contents and skills related to the area of Physical and Natural Sciences in each document. In the second stage, in order to verify the correspondence between them, we made the comparison across documents, selecting only the contents and skills of the Curricular Guidelines and the Essential Learnings that presented some correlation with the BFSL. For example, the contents and skills related to the theme "ecology" of the two curriculum documents were associated with the contents that addressed the same theme in the BFSL. The contents and skills that were not related to documents were not included in the items of the pilot instrument. The third and final stage consisted of the elimination of similar contents and skills, whose selection criterion took into account the PASEO competencies. This step was of great importance for the process of item development, as the presence of similar content and skills could generate equivalence between items so that one item could serve as a clue to the answer of another, which is not desired for an assessment instrument such as this.

By the end of this stage, we selected 60 contents and skills for the pilot test, 10 from the Curricular Guidelines for Physical and Natural Sciences and 50 from the Essential Learnings in Natural Sciences and Physical-Chemistry, of which 17 were for grade 7, 17 for grade 8 and 16 for grade 9.

### 2.2.3. Definition of the Representativeness of the Content

The representativeness of the content is characterised by the proportion with which each content or skill is reflected in the instrument [37]. In this sense, the 60 selected contents and skills were represented in 64 items. We emphasize that two skills of the Curricular Guidelines and two of the 8th grade Essential Learnings were used in the development of more than one item because they presented correspondence with more than one content of the BSFL.

Given the three dimensions of scientific literacy proposed by Miller [4], we distributed the representativeness of the ALCE content as follows: six items for the NOS dimension, seven for the ISTS dimension and 51 for the CS dimension. We detail the distribution of items for this last dimension in Table 2.

**Table 2.** Representativeness of the content of the CS dimension.

| Content | No. of Items | Content | No. of Items | Content | No. of Items |
|---|---|---|---|---|---|
| Environmental change | 3 | Atoms and chemical elements | 3 | Force, gravity and motion | 2 |
| Universe and solar system | 3 | Substances and Mixtures | 2 | Ecology | 4 |
| Internal geodynamics | 4 | Chemical reactions | 2 | Evolution | 1 |
| External geodynamics | 4 | Energy | 4 | Cells | 2 |
| Temperature and physical state changes of matter | 3 | Waves | 3 | Physiology | 11 |

Source: elaborated by the authors.

### 2.2.4. Drawing up the Specification Table

The initial specification table of the ALCE (Table 3) connects the dimensions of scientific literacy and the cognitive domains, indicating the number of items for each of them. As can be seen, of the 64 items, 26 belong to the cognitive domain of evaluate, 21 to understand and 17 to analyse.

**Table 3.** ALCE initial specification table.

| Scientific Literacy Dimension | Cognitive Domain | | | Total |
|---|---|---|---|---|
| | Understand | Analyse | Evaluate | |
| Nature of science (NOS) | 4 | 1 | 1 | 6 |
| Impact of science and technology on society (ISTS) | 2 | 1 | 4 | 7 |
| Content of science (CS) | 15 | 15 | 21 | 51 |
| Total | 21 | 17 | 26 | 64 |

Source: elaborated by the authors.

### 2.2.5. Construction of the Instrument

The construction of the instrument basically included the development of the items [35]. Therefore, this process involved making a decision about the format of the items, the technical guidelines and the configuration of the statements. We chose to use an adapted version of the true-false format, the *true-false-don't know*, with the purpose of reducing students' guesswork [48]. As technical guidelines, we used those proposed by [49], with detailed instructions for the elaboration of objective items, and by Ebel and Frisbie [48], specific for true-false items.

As for the configuration of the statements, we decided to use interpretative items, in which "students have to interpret, comprehend, analyze, apply, or synthesize the information presented" ([29], p. 146), as they meet the objectives of this instrument. We presented the information in text form, in which the statement is composed of one or more sentences highlighted in italics, accompanied by a statement, without highlighting, which should be answered as true or false, as shown in the example below:

*Imagine that an ice cube was left on top of the laboratory bench at an ambient temperature of 25 °C. After a few minutes it was noticed that there was water on the spot.* The physical explanation for this phenomenon is that the ice received heat from the environment, causing its molecules to move more freely, changing to a liquid state.

2.2.6. Theoretical Analysis of the Items

Considering the lack of a specific test for the theoretical analysis of the items, we performed a qualitative approach followed by a quantitative one [50]. In the qualitative approach, the items were submitted to a panel of experts in order to evaluate the correspondence between the item and the curriculum documents; the accuracy of the statements; the presence of ambiguities; the appropriateness of the language and vocabulary for the target audience; and the relevance of the item for scientific literacy. The panel was composed of four Elementary and/or Secondary School teachers, two of whom were in Natural Sciences and two in Physical Chemistry, and six Higher Education teachers, two of whom were in Educational Sciences, one in Biology, one in Geology, one in Physics and one in Chemistry.

As for the quantitative approach, we used the Content Validity Index (CVI), which "measures the proportion or percentage of judges who agree on certain aspects of the instrument and its items" ([50], p. 3065), in the relevance of the item for scientific literacy. We chose to eliminate the items that presented CVI lower than 0.8, which reduced the number of ALCE items to 35 (Table 4).

**Table 4.** Pilot instrument specification table.

| Content Dimension | Cognitive Domain | | | Total |
|---|---|---|---|---|
| | Understand | Analyse | Evaluate | |
| Nature of science (NOS) | 4 | 1 | 1 | 6 |
| Impact of science and technology on society (ISTS) | 1 | 1 | 4 | 6 |
| Content of science (CS) | 5 | 9 | 9 | 23 |
| Total | 10 | 11 | 14 | 35 |

2.2.7. Empirical Analysis of the Items

The empirical analysis was performed using data from the pilot test, which was answered by 176 students in the 10th grade from eight schools in the southern region of mainland Portugal, at the beginning of the 2020/2021 school year. As a result of the quarantine due to COVID-19 and with the purpose of reaching the largest number of schools, obtaining as many answers as possible, the pilot test was answered in an online format, through the LimeSurvey software, in the classroom and in the presence of the teachers, who presented the study procedures to the students, administered and monitored the application of the instrument. All consent procedures and interactions with the study subjects were previously authorised by the Directorate-General for Innovation and Curricular Development (DGICD), Monitoring of Surveys in the School Environment (MIME), under registration no. 0740900001. The maximum response time was 50 min. It is worth mentioning that the students did not receive any special preparation in scientific literacy besides the approach performed in the science subjects' classes.

Traditionally, in the empirical analysis of the items, psychometrics analyzes the indices of difficulty and discrimination of the items and the rate of guessing at random [35]. However, our study analysed only the difficulty parameter of the items, since guessing was minimised by the answer option *don't know* and the discrimination parameter, whose aim is to analyse the ability of an item to "differentiate subjects with different magnitudes of trait of which the item constitutes the behavioural representation" ([35], p. 139), is more important for large-scale tests and not for pedagogical assessments [51], as is the case of ALCE, which purpose is to provide information to teachers, in order for them to reflect on the students' performance regarding the contents that are being assessed.

To calculate the index of difficulty, we used the two-parameter logistic model of Item Response Theory (IRT), model that best fit the data according to the analysis of variance performed ($p < 0.5$), through the software RStudio, in its version 3.6.0. We justify the choice for IRT by the fact that it analyzes the items individually, without the total scores of the

test directly influence the analysis [52–54]. As a criterion for classifying the difficulty index of the items, we used the categorization proposed by Ferreira [55], which was based on Baker's [53] categories (Table 5).

**Table 5.** Range of values of the levels of difficulty of the items.

| Level | Range of Values |
|---|---|
| Very easy | $\leq -1.28$ |
| Easy | $-1.27--0.52$ |
| Medium | $-0.51$–$0.51$ |
| Difficult | $0.52$–$1.27$ |
| Very difficult | $\geq 1.28$ |

Source: Elaborated by the authors.

After applying the pilot test, we observed that the level of difficulty of the items of the instrument was distributed as follows: seven very easy items; six easy items; ten medium items; four difficult items; and eight very difficult items. The distribution of the difficulty level of each subtest and of the instrument is presented in Table 6. Table 7 identifies the index and the difficulty level of each item of the pilot instrument.

**Table 6.** Number and percentage of items of each subtest and the pilot instrument by difficulty level.

| | NOS | ISTS | CS | ALCE (Pilot) |
|---|---|---|---|---|
| | No. of Items (%) | No. of Items (%) | No. of Items (%) | No. of Items (%) |
| Very easy | 3 (50) | 2 (33) | 2 (9) | 7 (20) |
| Easy | 0 (0) | 1 (17) | 5 (22) | 6 (17) |
| Medium | 2 (33) | 2 (33) | 6 (26) | 10 (29) |
| Difficult | 1 (17) | 0 (0) | 3 (13) | 4 (11) |
| Very difficult | 0 (0) | 1 (17) | 7 (30) | 8 (23) |

Source: Elaborated by the authors.

**Table 7.** Indexes and levels of difficulty of the items of the pilot instrument.

| | NOS | | | CS | |
|---|---|---|---|---|---|
| **Item** | **Index (b)** | **Level** | **Item** | **Index (b)** | **Level** |
| 1 | 0.81 | D | 17 | −2.01 | VE |
| 2 | −1.83 | VE | 18 | −2.88 | VE |
| 3 | −2.17 | VE | 19 | 0.27 | M |
| 4 | −4.01 | VE | 20 | −0.41 | M |
| 5 | −0.45 | M | 21 | 4.88 | VD |
| 6 | −0.11 | M | 22 | 0.72 | D |
| | | | 23 | 3.91 | VD |
| | **ISTS** | | 24 | 5.45 | VD |
| 7 | −1.29 | VE | 25 | −0.76 | E |
| 8 | 0.06 | M | 26 | 1.08 | D |
| 9 | −1.45 | VE | 27 | 9.66 | VD |
| 10 | −0.47 | M | 28 | −0.56 | E |
| 11 | 4.11 | VD | 29 | 1.77 | VD |
| 12 | −1.23 | E | 30 | −0.17 | M |
| | | | 31 | 2.52 | VD |
| | **CS** | | 32 | 1.20 | D |
| 13 | −0.96 | E | 33 | −0.73 | E |
| 14 | −0.52 | E | 34 | −0.25 | M |
| 15 | 0.28 | M | 35 | −0.43 | M |
| 16 | 1.58 | VD | | | |

Note: VE = very easy; E = easy; M = medium; D = difficult; VD = very difficult. Source: Elaborated by the authors.

Analysing the results of the pilot test as a whole, it can be seen that the level of difficulty of the items is distributed approximately homogeneously, with the exception of the number of difficult items, which is slightly below the other levels. Calculating the average difficulty level of each subtest and the instrument, we identify the subtest of the NOS as easy and the subtests of the ISTS, the CS and the instrument as medium.

However, the analysis by subtest reveals the heterogeneity across subtests, given that the NOS subtest has the highest percentage of very easy items, and the CS subtest has the highest percentage of very difficult items. Therefore, the analysis performed pointed out the need to revise these items and the possible elimination of those that do not present good technical quality.

### 2.3. Gathering Validity Evidence Based on Internal Structure

The data gathered in the pilot test were also used to collect validity evidence based on internal structure of the ALCE. In accordance with what is established by the Standards [33], the types of analysis applied in this study were in accordance with the use for which this instrument is being proposed, which can be summarised as a pedagogical assessment aimed at providing information to teachers about the students' performance in scientific literacy, so that they have data to enable them to reflect on their practice and modify it in order for students to finish the 3rd cycle of Basic Education and be able to use the knowledge and skills acquired to solve and explain everyday problems.

As well as in the gathering of evidence based on content, we used Pasquali's [37] framework to gather validity evidence based on internal structure. According to the author, these can be designed, for example, by analysing the behavioural representation of the construct, the analysis by hypothesis or the IRT information function.

In the present study, we chose to use the IRT information function. We justify our choice due to the fact that the IRT information function, as a strategy for gathering validity evidence based on internal structure, presents itself as an index of instrument accuracy [34, 54,56,57], capable of assessing the strength or salience of the primary aspects underlying an assessment, which is linked to the validity of internal consistency [34].

The IRT information function can be expressed through the information curve, a graphical representation that reveals for which range or ranges of proficiency levels, or latent trait, the instrument is particularly valid and for which is not [57]. The information function can also be represented by means of the standard error of measurement, called standard error of estimation in IRT, which is characterised by the inverse of the information curve of the test and allows the analysis of the accuracy of the assessment instrument [57–59].

Consequently, we resorted to the software RStudio, in its version 3.6.0, to perform the information function analysis of the subtests. We used the same resource to analyse the students' proficiency level ($\theta$) and the Kernel density estimate, which analyses the density of students as a function of $\theta$.

The analysis of the subtest information function was designed through the (sub)test information curves (TIC) and the calculation of the amount of total information available in each subtest. According to Reeve and Fayers [60] and Baker and Kim [58], the TIC is influenced by the difficulty and discrimination parameters of the items. While the difficulty index influences the location of the information curve on the horizontal axis of $\theta$, the discrimination index acts on the vertical axis, the height of the curve [58,60]. The wider the curve, the greater the amount of $\theta$ levels covered by the test information. The higher the curve, the greater the amount of information for that specific $\theta$.

For Baker and Kim [58], the ideal information function should be a horizontal straight line over the widest possible area, which would reveal that all information includes as much $\theta$ as possible. However, such a situation is very unlikely to occur and consequently, "test information function should be rounded in appearance over the ability range of most interest" ([58], p. 163). The results of this analysis showed that the highest amount of information and, consequently, highest accuracy and lowest standard error of estimation in

the NOS and ISTS subtests come from the range of θ between −1 and 1. In the CS subtest, this range lies from −3.0 to 1.0 on the scale of θ (Figure 1).

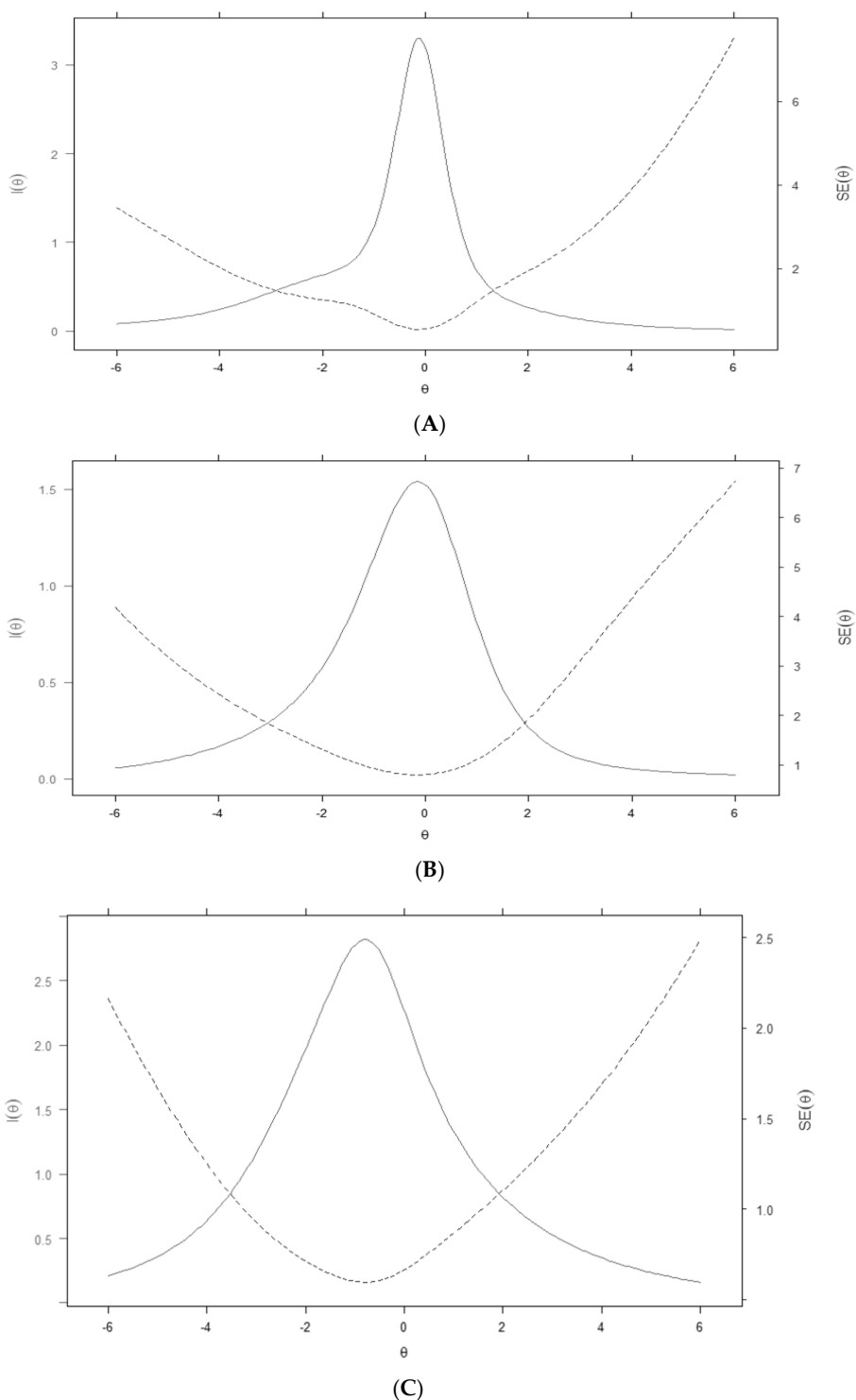

**Figure 1.** Test information curve and standard error of measurement per subtest of the pilot instrument. Notes: (**A**) NOS subtest; (**B**) ISTS subtest; (**C**) CS subtest; I(θ) = amount of subtest information; θ = profi-ciency scale; SE(θ) = standard error; Continuous line (—) = I(θ); and Dashed line (– – –) = SE(θ). Source: elaborated by the authors.

Analysis of the total amount of information for each subtest indicated that the subtests of the NOS, the ISTS and the CS produce 7.11, 5.27 and 13.40 of information, respectively

(Table 8). Considering the typical proficiency range, θ between −3 and 3 (Baker and Kim, 2017), the amount of information was 6.11 for the NOS subtest, 4.54 for the ISTS subtest and 10.09 for the CS subtest.

**Table 8.** Amount of information from each subtest, the pilot test.

| Subtest | No. of Items | Total Amount of Information | Amount of Information (%) in the Range of θ from −3 to 3 |
|---|---|---|---|
| NOS | 6 | 7.11 | 6.11 (86.04%) |
| ISTS | 6 | 5.27 | 4.54 (86.20%) |
| CS | 23 | 13.40 | 10.09 (75.28%) |

Source: elaborated by the authors.

The results of the level of θ of the students evidenced that the average θ of the students in the three subtests is located in the value zero, in the centre of the proficiency scale (Table 9). Kernel density estimation analysis, on the other hand, revealed that most of the students are concentrated in the range of θ between −1 and 1 (Figure 2).

**Table 9.** Average level of θ of students per subtest of the pilot instrument.

| | θ | | |
|---|---|---|---|
| | NOS | ISTS | CS |
| M | 0.00 | 0.00 | 0.00 |
| SD | 0.78 | 0.72 | 0.79 |
| Maximum | 1.06 | 1.05 | 2.03 |
| Minimum | −1.70 | −1.48 | −2.36 |

Note. θ = students' proficiency; M = mean; and SD = standard deviation. Source: elaborated by the authors.

Comparing the results of the three analyses (information function, level of θ of the students and the Kernel density estimate), we found that, in the case of the NC and ICTS subtests, the vertex of the information and density curves correspond to the interval where the average level of θ presented by the students is located, zero. In the subtest of the CS, we find a small difference because, while most of the information is located in the interval between −3 and 1 on the ability scale, the average θ of the students is equal to zero, and the highest density is also close to θ and equal to zero. Therefore, we infer that the items of the NOS and ISTS subtests were the ones that obtained the highest accuracy and best capability to assess the students' scientific literacy skills and that the items of the CC subtest, despite being a little less accurate, are still capable of assessing them and gathering valid information about the students' scientific literacy.

Moreover, analysing the amount of information of each subtest, which depends, in part, on the number of items that compose them [58,61], we found that the CS subtest—which has almost four times the number of tens of the other subtests—is the one with the highest amount of information. . When analysed only the typical range of θ for normal tests (between −3 and 3) [58], all subtests are able to provide above 75% of the total information of the test.

Likely, the difference in the results of the items of the subtest of the CS is associated with the difficulty index of the items that compose it, since it should be located around the midpoint of the θ range of interest [58]. As shown in the gathering of validity evidence based on content, 43% of the items classified as difficult or very difficult belong the subtest of the CS, contrasting with the 17% of the subtests of the NOS and the ISTS.

Hence, the collections of validity evidence based on content and internal structure of the ALCE pilot test application partially support the use of the results of the items of the three subtests for the assessment of students' scientific literacy. However, we acknowledge that the items classified as very easy and very difficult should be revised in order to better fit the characteristics of the assessed sample.

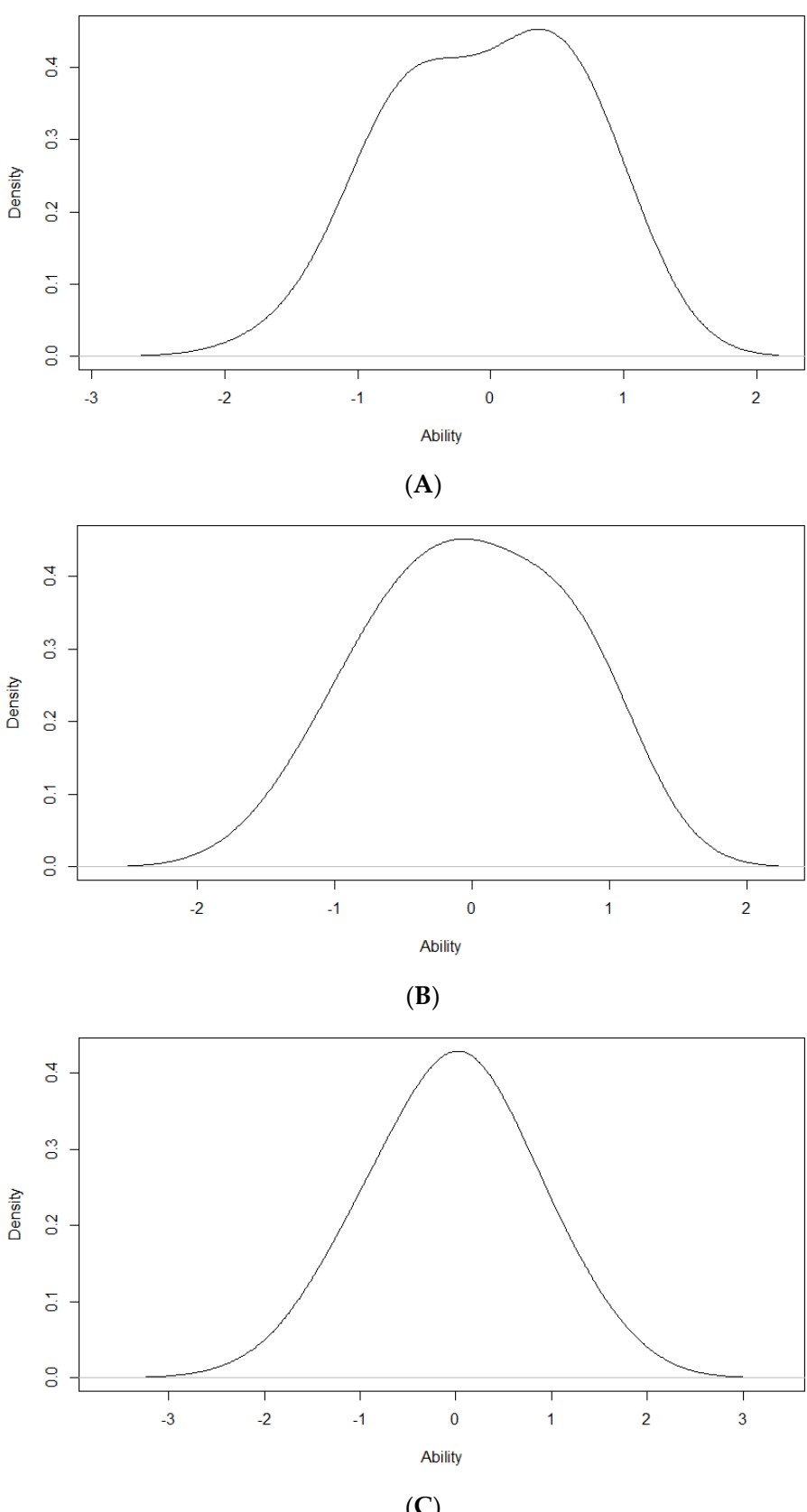

**Figure 2.** Kernel density estimation according to the students' θ in each subtest of the pilot instrument. Notes: (**A**) Subtest of the NOS; (**B**) subtest of the ISTS; and (**C**) subtest of the CS. Source: elaborated by the authors.

*2.4. Review of the Items after the Pilot Test*

Aiming to develop an instrument to be used by teachers to assess the scientific literacy level of students at the end of the 3rd cycle of Basic Education, the items of the ALCE needed to present a difficulty level minimally consistent with the level of $\theta$ of the students. Thus, all items with difficulty index higher than 0.75 (items 1, 11, 16, 21, 22, 23, 24, 26, 27, 31 and 32) were forwarded to the panel of experts (on this occasion, only for the four teachers of the 3rd cycle of Basic Education and/or Secondary Education, given that they are the experts who are in daily contact with the students), who answered the following questions: (1) In your opinion, why did the students feel difficulty on this item? (2) Taking into account that the instrument intends to assess scientific literacy, would you eliminate this item? (3) What changes would you suggest to decrease the difficulty of this item?

The experts' feedback was essential to ensure that the items collected valid and appropriate information about the students' level of scientific literacy. We used the comments and answers to the three questions to make changes to the level of comprehension, relevance and readability of the items and to determine whether those items should be excluded from the final version of the instrument.

We also made minor changes to the items that raised greater doubts on the part of the students (items 2, 5, 7, 8, 10, 15, 16, 30, 34 and 35). We collected this information through a statement item added at the end of the set of items of each subtest of the pilot instrument, which asked students whether any item raised doubts of content, vocabulary, or other nature, for which students should indicate the item number and their respective doubt. The information collected allowed us to change words and expressions that raised doubts in students and may have interfered with their answers to the items.

Finally, we reviewed the items with difficulty level below $-0.75$ (items 2, 3, 4, 7, 9, 12, 13, 17, 18 and 25) in order to identify possible answer facilitators. We changed the configuration or the words used of the items 2 and 7. We found no problems with the remaining items, for which we understand that the ease of the item is due to the fact that students mastered their respective contents and skills.

This set of revisions, which guided the development of the final instrument, made it possible to identify and solve failures in 21 items and eliminate 1 item from the CS subtest, more specifically from the area of Chemistry. As a result, the final version of the ALCE consists of 34 items, according to the specification table presented in Table 10.

**Table 10.** ALCE specification table by dimension of the content/cognitive domain.

| Content Dimension | Cognitive Domain | | | Total |
|---|---|---|---|---|
| | Understand | Analyse | Evaluate | |
| Nature of science (NOS) | 4 | 1 | 1 | 6 |
| Impact of science and technology on society (ISTS) | 1 | 1 | 4 | 6 |
| Content of science (CS) | 5 | 8 | 9 | 22 |
| Total | 10 | 10 | 14 | 34 |

Source: elaborated by the authors.

*2.5. Application of the ALCE*

The ALCE final application took place in the period from April to June 2022, at the end of the 2021/2022 school year. The test was answered by 516 students in the 9th grade from 20 public schools in mainland Portugal, of which 259 (50.2%) were male and 257 (49.8%) were female. The average age of the respondents was 14.69 years (SD = 0.88).

The same statistical method used to gather validity evidence for the pilot test, the IRT two-parameter logistic model, was employed to gather such information from the final version of the ALCE. For the gathering of validity evidence based on test content, we conducted once again the empirical analysis of the items and, for the gathering of evidence of validity based on internal structure, we resorted to the analyses of the information function, students' level of $\theta$ and Kernel density estimation.

The results of the empirical analysis of the ALCE items identified the following distribution of the level of difficulty of the items: five very easy; eight easy; seven medium; six difficult; and eight very difficult. The distribution of the difficulty level of each subtest and of the instrument is presented in Table 11, while Table 12 exposes the index and the difficulty level of each item of the ALCE.

**Table 11.** Number and percentage of items of each subtest and of the ALCE by difficulty level.

| Level | NOS | ISTS | CS | ALCE |
|---|---|---|---|---|
| | No. of Items (%) | No. of Items (%) | No. of Items (%) | No. of Items (%) |
| Very easy | 2 (33) | 1 (17) | 2 (9) | 5 (14) |
| Easy | 1 (17) | 3 (50) | 4 (17) | 8 (23) |
| Medium | 2 (33) | 0 (0) | 5 (22) | 7 (20) |
| Difficult | 1 (17) | 2 (33) | 3 (13) | 6 (17) |
| Very difficult | 0 (0) | 0 (0) | 8 (35) | 8 (23) |

Source: elaborated by the authors.

**Table 12.** Indexes and levels of difficulty of the items of the ALCE.

| NOS | | | CS | | |
|---|---|---|---|---|---|
| Item | Index (b) | Level | Item | Index (b) | Level |
| 1 | 0.96 | D | 16 | 1.23 | D |
| 2 | −0.91 | E | 17 | −1.42 | VE |
| 3 | −1.50 | VE | 18 | −2.00 | VE |
| 4 | −1.31 | VE | 19 | −0.14 | M |
| 5 | −0.03 | M | 20 | 5.78 | VD |
| 6 | −0.40 | M | 21 | 1.10 | D |
| | | | 22 | 7.19 | VD |
| | **ISTS** | | 23 | 7.06 | VD |
| 7 | −1.15 | E | 24 | −0.52 | E |
| 8 | 0.58 | D | 25 | 2.76 | VD |
| 9 | −1.65 | VE | 26 | 4.24 | VD |
| 10 | −0.94 | E | 27 | −0.17 | M |
| 11 | 0.58 | D | 28 | 1.80 | VD |
| 12 | −0.79 | E | 29 | −0.42 | M |
| | | | 30 | 1.57 | VD |
| | **CC** | | 31 | 5.11 | VD |
| 13 | −0.80 | E | 32 | −0.80 | E |
| 14 | −0.59 | E | 33 | 0.29 | M |
| 15 | 0.57 | D | 34 | −0.14 | M |

Note: VE = very easy; E = easy; M = medium; D = difficult; VD = very difficult. Source: elaborated by the authors.

We noticed a slight increase in the homogeneity in the distribution of the level of difficulty of the items. However, we found that while the average difficulty level of the subtests of the NOS and the ISTS remained, easy and medium, respectively, the average difficulty level of the subtest of the CS, which came to include all items classified as very difficult, rose to the difficult level.

In a qualitative analysis of the eight items that obtained a very difficult classification, we found that five of them belong to the subject of Physical Chemistry (items 20, 22, 23, 28, 30 and 31) and three to the subject of Natural Sciences (items 28, 30 and 31). Of these, three are in the cognitive domain of understanding (22, 23 and 31), two in the domain of analysis (23 and 25) and three in the cognitive domain of evaluation (20, 26 and 28). This fact is consistent with the results of Portuguese students in the 2019 Trends in International Mathematics and Science Study (TIMSS) assessment, which found that, despite demonstrating some knowledge in the areas of Biology and Physics, few students are able to apply the knowledge and skills of Biology, Chemistry, Physics and Earth Sciences

and characterize the concepts of such areas in a contextual plurality [62]. They also are compatible with the results presented in the last report of the Programme for International Student Assessment (PISA) of Portugal, which revealed that the vast majority of students did not achieve the proficiency level in which they have to "apply their knowledge of and about science autonomously and creatively to a wide variety of situations, even the less familiar ones" ([63], p. vii).

Regarding the analysis of the information function of the final version of the ALCE, we observed a slight flattening of the curves of the three subtests when compared to those of the pilot test, increasing the amount of information in the θ of the extremities (Figure 3). In the NOS subtest, for example, the highest amount of information and accuracy and the lowest standard error were now in the range of θ between −1.5 and 1.5. In the ISTS subtest, the largest amount of information remained in the interval between −1 and 1, but there was an increase in the amount of information at the extremities. Finally, in the CS subtest, the interval with more information became between θ of −3 and 2.

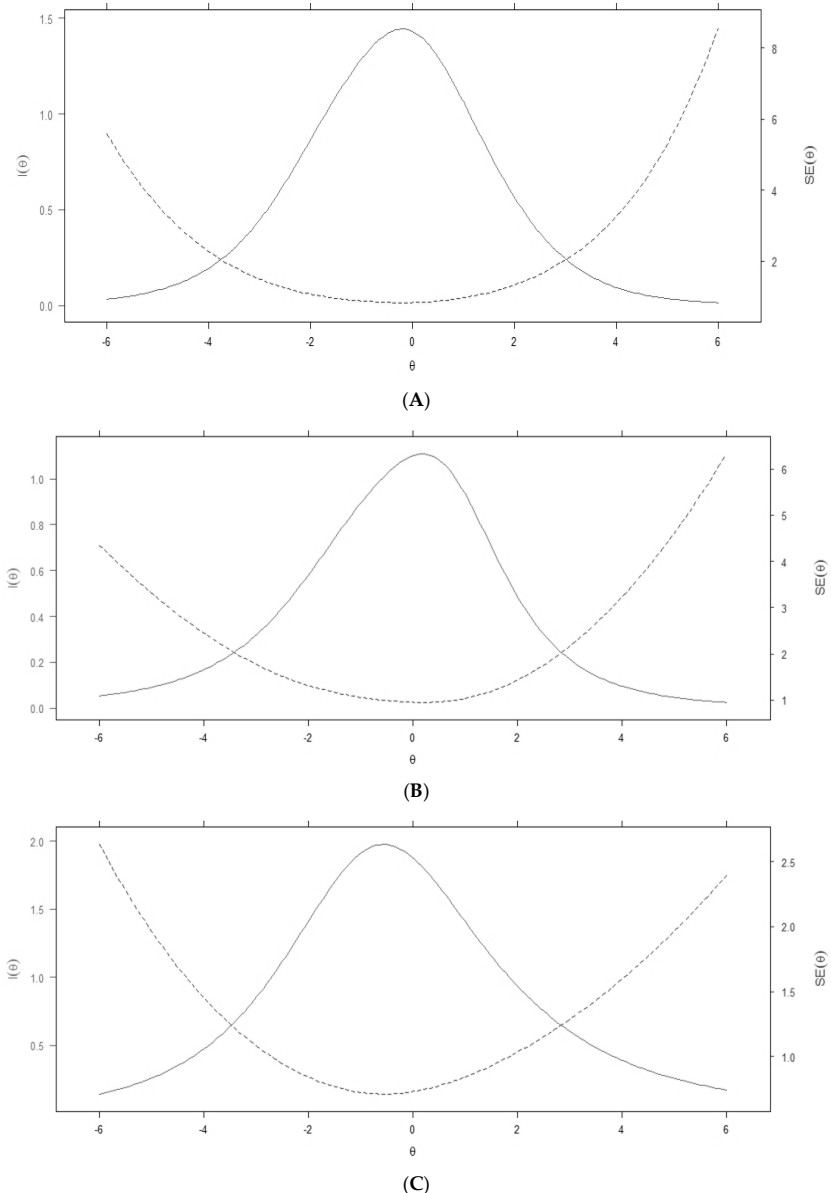

**Figure 3.** Test information curve and standard error of measurement per subtest of the ALCE. Notes: (**A**) NOS subtest; (**B**) ISTS subtest; (**C**) CS subtest; I(θ) = amount of subtest information; θ = proficiency scale; SE(θ) = standard error; Continuous line (—) = I(θ); Dashed line (− − −) = SE(θ). Source: Elaborated by the authors.

This slight flattening in the curves is confirmed by analysing the total amount of information from the ALCE subtests. The results indicated that the subtests of the NOS, the ISTS and the CS produce 6.38, 5.11 and 11.17 of information, respectively (Table 13). Considering the typical proficiency range, θ between −3 and 3 [58], the amount of information was 5.60 for the NOS subtest, 4.30 for the ISTS subtest and 8.33 for the CS subtest.

**Table 13.** Amount of information from each subtest of the ALCE.

| Subtest | No. of Items | Total Amount of Information | Amount of Information (%) in the Range of θ from −3 to 3 |
|---|---|---|---|
| NOS | 6 | 6.38 | 5.60 (87.81%) |
| ISTS | 6 | 5.11 | 4.30 (84.06%) |
| CS | 22 | 11.17 | 8.33 (74.60%) |

Source: elaborated by the authors.

Therefore, a small decrease in the amount of information in the three subtests can be observed, being the ISTS the one that obtained the slightest reduction. This was also the subtest that presented the greatest observable difference in the information curve. While in the pilot test the peak of the curve was located in the negative part of the θ scale, in the final application it was located in the positive part. This result may be associated with the better adequacy of the difficulty index of the items of this subtest to the level of θ of the students [58].

Regarding the level of θ of the students and the Kernel density estimate, the results of the final ALCE application resemble those of the pilot test. The average level of θ of the students in the three subtests was located at the value zero (Table 14), and most of the students concentrated in the θ range between −1 and 1 (Figure 4).

**Table 14.** Average level of θ of students per subtest of the ALCE.

| | Θ | | |
|---|---|---|---|
| | NOS | ISTS | CS |
| M | 0.00 | 0.00 | 0.00 |
| SD | 0.74 | 0.69 | 0.78 |
| Maximum | 1.29 | 1.24 | 2.19 |
| Minimum | −1.63 | −1.45 | −2.03 |

Note. Θ = students' proficiency; M = mean; and SD = standard deviation. Source: elaborated by the authors.

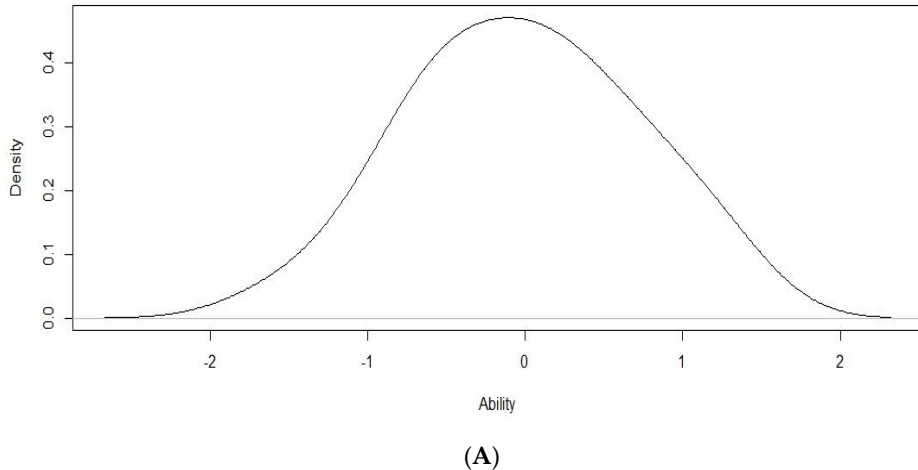

(**A**)

**Figure 4.** *Cont.*

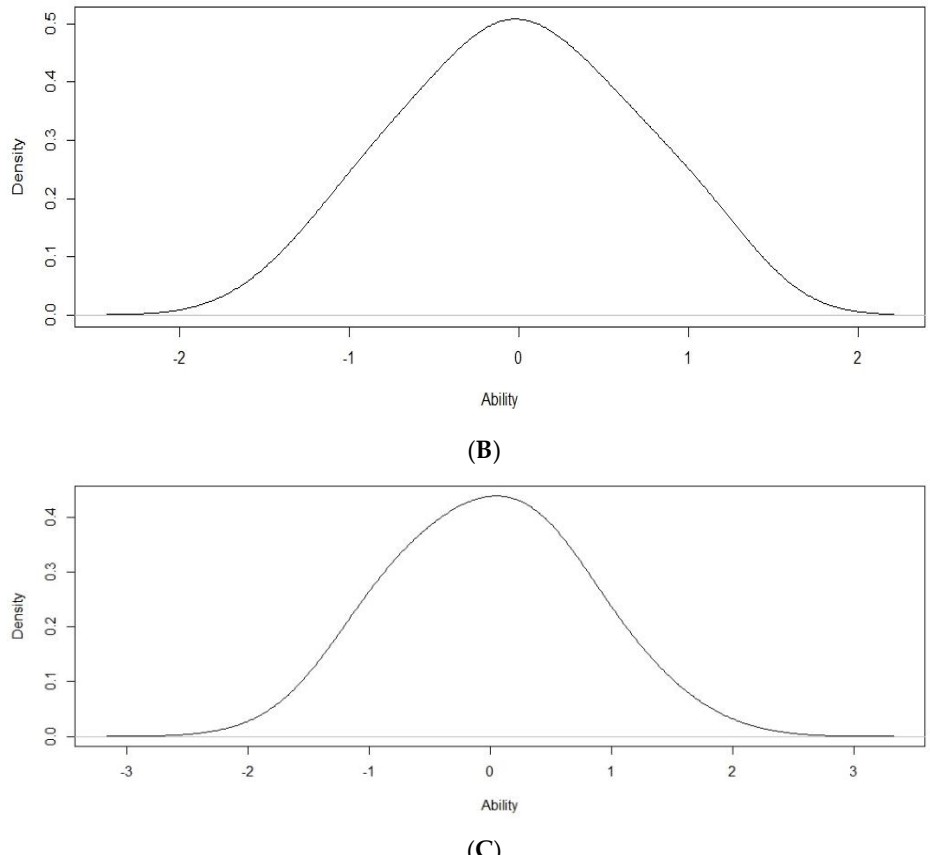

**Figure 4.** Kernel density estimation according to the students' θ in each subtest of the pilot instrument. Notes: (**A**) Subtest of the NOS; (**B**) subtest of the ISTS; and (**C**) subtest of the CS. Source: elaborated by the authors.

When comparing the distribution of the level of θ of the students in percentage, in the pilot test and in the final application (Table 15) and according to the levels of difficulty stipulated for this study, we observe that the results are similar. Highlights are given to the subtests of NOS and CS that, in the final application, increased by 10% and 6%, respectively, of students with θ equal to or greater than 1.28. On the other hand, in the ISTS subtest, a 7% decrease was registered in the number of students with θ between 0.52 and 1.27.

**Table 15.** Distribution of students' θ between the pilot test and the final application, by subtest.

| θ | NC | | | ICTS | | | CC | | |
|---|---|---|---|---|---|---|---|---|---|
| | **Pilot** | **Final** | ≠ | **Pilot** | **Final** | ≠ | **Pilot** | **Final** | ≠ |
| ≤−1.28 | 5% | 4% | −1% | 3% | 2% | −1% | 13% | 4% | −9% |
| −1.27−−0.52 | 24% | 24% | 0 | 21% | 22% | +1% | 18% | 23% | +5% |
| −0.51−0.51 | 42% | 46% | +4% | 45% | 52% | +7% | 50% | 47% | −3% |
| 0.52−1.27 | 29% | 16% | −13% | 31% | 24% | −7% | 20% | 21% | +1% |
| ≥1.28 | 0% | 10% | +10% | 0% | 0% | 0% | 0% | 6% | +6% |

Note. θ = students' proficiency level; ≠ = difference. Source: elaborated by the authors.

### 2.6. Scoring and Classification

We have opted for assigning the score of the instrument in a dichotomous manner: items answered correctly received one point, and items answered incorrectly and marked with the option *don't know* received zero points. For the categorization, we determined five levels of scientific literacy (very low, low, moderate, high and very high) for each subtest, according to the percentage and number of correct answers in the items (Table 16).

**Table 16.** Levels of scientific literacy according to the percentage and number of correct answers per subtest.

| Levels | Percentage | NOS | ISTS | CS |
|---|---|---|---|---|
| 1—Very low | <20% | $\leq 1$ | $\leq 1$ | $\leq 4$ |
| 2—Low | 20–49% | 2 | 2 | 5–10 |
| 3—Moderate | 50–69% | 3–4 | 3–4 | 11–15 |
| 4—High | 70–89% | 5 | 5 | 16–19 |
| 5—Very high | $\geq 90\%$ | 6 | 6 | $\geq 20$ |

Source: elaborated by the authors.

This distribution follows the assessment scoring model currently adopted in the 3rd cycle of Portuguese Basic Education [64], whereby tests are graded on a scale from 0% to 100%, and their final classification is further converted into a scale from 1 to 5, according to the following distribution: level 1, from 0 to 19%; level 2, from 20 to 49%; level 3, from 50 to 69%; level 4, from 70 to 89%; and level 5, from 90 to 100%. To be considered approved, students must reach at least level 3 on the scale.

For the general classification of the students' level of scientific literacy, we stipulated the same levels, established by calculating the average of the scientific literacy levels of the three subtests (Table 17). Considering that the definition of scientific literacy does not imply an ideal, or even acceptable, level of understanding but rather a minimum level [4], and for the minimum level (level 3) for students to pass the subjects of the Portuguese 3rd cycle of Basic Education [64], we established that students who achieve at least a moderate level of scientific literacy are considered scientifically literate.

**Table 17.** General scientific literacy levels according to the average of the subtest levels.

| Levels | Average of the Subtest Levels |
|---|---|
| Very low | <1.00 |
| Low | 1.00–2.49 |
| Moderate | 2.5–3.49 |
| High | 3.5–4.49 |
| Very high | $\geq 4.5$ |

Source: elaborated by the authors.

### 2.7. Results of the ALCE Final Application

The results obtained from the final application of the ALCE [65] revealed that of the 516 responding students, 184 (35.7%) were at the low level, 255 (49.4%) were at the moderate level, 74 (14.3%) were at the high level, and 3 (0.6%) were at the very high level (Figure 5). There were no students classified at the very low level of scientific literacy.

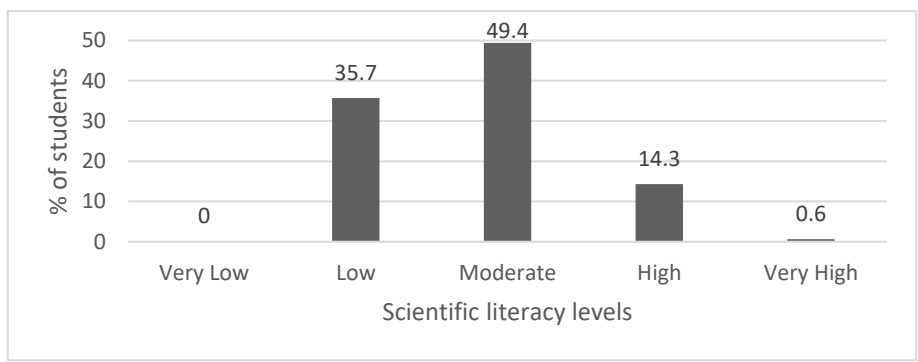

**Figure 5.** Students' scientific literacy level. Source: elaborated by the authors.

In concordance with the ALCE classification rationale, we found that 64.14% of students were classified as scientifically literate. We also found that only 14.9% surpassed the

moderate level of scientific literacy and that 35.7% of pupils did not achieve the minimum number of correct answers on the subtests to be classified as scientifically literate.

Although it is not possible to make a rigorous and direct comparison between the grade levels, these results are in accordance with those disclosed in the latest reports of Portuguese students' performance in TIMSS 2019 [62] and PISA 2018 [63] that are available. In the TIMSS 2019 Science test, 73% of 8th grade students reached the intermediate level, 34% reached the high level and only 7% reached the advanced level of performance on the TIMSS scale [62]. In the PISA 2018 Science test, while 80% of students reached at least level 2 of proficiency on the PISA scale, only 5.1% and 0.5% of students reached levels 5 and 6, respectively, of the scale [63].

Analysing the results by subtest, we identify that most students are at the moderate level of scientific literacy on the NOS and ISTS subtests, 44.8% and 49.2%, respectively, and at the low level on the CS subtest, 55.63% (Figure 6). Looking at the results of the subtests separately and considering the ALCE classification rationale, that it is necessary to achieve at least a moderate level of scientific literacy, 70% of the students can be considered scientifically literate in the NOS subtest, 72.9% in the ISTS subtest and 39.9% in the CS subtest.

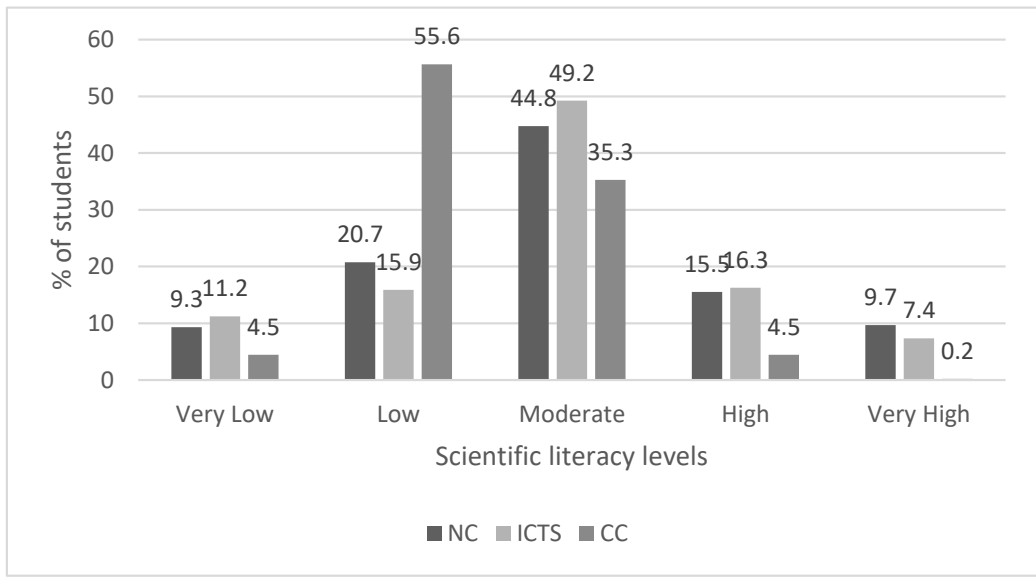

**Figure 6.** Distribution of students' scientific literacy level by subtest. Source: elaborated by the authors.

Overall, the results indicate that the students' performances were higher in the items whose knowledge and skills are related to the basic foundations of science, consisting of the scientific perception of natural phenomena, the methodology used in research and the very nature of the scientific enterprise, which means the nature of science (NOS) and the understanding of the risks and opportunities of using scientific-technological knowledge, achievements and products for society, that is to say the impact of science and technology on society (ISTS), and lower in the items assessing aspects related to the vocabulary, terms, expressions and basic content of natural sciences and Physical Chemistry, represented by the content of science (CS).

Once again, these results resemble those of the TIMSS and PISA tests. According to the TIMSS 2019 report, few Portuguese students have the skills to apply the knowledge acquired in the subjects of Biology, Chemistry, Physics and Earth Sciences and characterize their concepts in a contextual plurality [62]. Whereas in the PISA 2018 report, the authors state that only 5.6% of students are able to apply scientific knowledge autonomously and creatively to a variety of situations, including those that are less familiar to them.

## 3. Implications and Conclusions

Responding to the need for the elaboration of new instruments for the assessment of scientific literacy, especially those designed for primary school students, as mentioned by Coppi et al. [18], we developed an instrument for the assessment of scientific literacy aimed at students at the end of the 3rd cycle of Basic Education. The ALCE is an instrument composed of *true-false-don't know* items, developed to be administered in a period of one lesson (50 min) to 9th-grade classes, at the end of the school year as a summative assessment of learning, or even to 10th-grade classes, at the beginning of the school year as a diagnostic assessment.

The ALCE was not designed to assess factual knowledge, such as photosynthesis, the laws of thermodynamics or atomic structure but rather to assess the ability to use the knowledge and skills, present in the main curriculum documents in the area of Physical and Natural Sciences and that should be developed in the subjects of Natural Sciences and Physical Chemistry, for the explanation or resolution of everyday phenomena and situations. The development of the instrument was based on the current literature on the construction of assessment instruments and used the gathering of evidence based on content and internal structure to determine the validity of the use of its results.

Theoretically, the ALCE provides a valid set of information about students' scientific literacy levels. The validity evidence gathered in this study, in alignment with that established by the Standards [33], demonstrates that the results of applying the ALCE can be used for decision making. Furthermore, the results of our process of item construction, administration and review provide support that the ALCE is well aligned with the underlying conceptualisation of scientific literacy that we seek to assess.

In this study, we have employed the ALCE to assess 516 students from 20 schools in mainland Portugal and identified that most of them (49.4%) are at the moderate level of scientific literacy and that 64.14% were classified as scientifically literate. The results also revealed that 70% of the students were considered scientifically literate in the NOS subtest, 72.9% in the ISTS subtest and 39.9% in the CS subtest, the latter being the subtest in which students encountered the major difficulties.

Taking into account the characteristics evidenced by ALCE, our proposal is that it should be used by science teachers with their students so that they can use the results to identify the dimensions, contents and scientific literacy competencies with more and less potential and the gap between the teaching objectives of the subjects and the students' proficiency. Such results might also be able to enable reflection on their own objectives, methodologies, lesson plans and strategies used in the classroom in order to modify them to better develop students' scientific literacy. The ALCE is also informative in the sense that it is able to reveal misunderstood or misinterpreted competencies by means of the *don't know* response options which, if widely marked, may represent cases whose content and skills deserve teachers' attention.

Moreover, we envisage that ALCE might be useful for other purposes, including evaluating the Natural Sciences and Physical Chemistry curricula; providing indicators to help monitor the progress of science education at regional and national levels (if applied on a large scale, in a measurement perspective); and even serving as an auxiliary tool and a framework for teachers and researchers to develop their own instruments to assess students' scientific literacy.

We encourage teachers and researchers interested in using ALCE to contact the corresponding authors with requests for additional information. We also welcome feedback regarding findings and comments for revisions to future versions of ALCE.

## 4. Limitations

Firstly, our sample, despite involving 516 students, was restricted to 20 schools in mainland Portugal, not covering the administrative regions of the Archipelago of Madeira and the Archipelago of the Azores. This fact, also associated with the number of students in the sample, limits the generalisation of the results of this research to a national scale.

We suggest that future research should include these regions and different regional and cultural contexts in order to gather more information regarding the literacy level of students at the end of the 3rd cycle of Basic Education in Portugal. We point out, however, that the participating schools represent a variety of regions and socioeconomic levels.

Secondly, we provide validity evidence based on content and internal structure of the instrument, not referring to that based on response processes, other variables, and the consequences of the assessments. We reason our choice of these two types of validity evidence by the fact that, according to the Standards, it is not always necessary to collect all five types of validity evidence for all validation processes of assessment instruments, but "rather, support is needed for each proposition that underlies a proposed test interpretation for a specified use" ([33], p. 14). Furthermore, we consider the fact that evidence based on content is the primary source of evidence for an assessment instrument [66] and that the collection of evidence-based on internal structure is considered one of the fundamental ways to analyse the validation process related to assessment instruments, "since it is the direct way to verify the hypothesized legitimacy of the behavioral representation of latent traits" ([35], p. 996).

Nevertheless, we suggest that future research might be able to, for example, establish relationships between ALCE scores with other criteria, such as students' grades in the subjects of Natural Sciences and Physical Chemistry or scores on a similar test, in order to gather evidence based on relations to other variables. Evidence based on the response process could also be obtained, for example, from a larger sample of teachers on the expert panel. Although we shared the initial version and items, which after pilot testing, were contestable with four teachers (two from Natural Sciences and two from Physical Chemistry), a review by a larger group of teachers could be helpful in moving forward with revised versions of the ALCE. The collection of these two types of evidence would further legitimize the widespread use of this instrument.

Finally, third, although the validity evidence collected enables and attests the use of the ALCE and, consequently, its results, we recognize that some of its items, due to the high difficulty index, did not prove to be fully adequate for the proficiency level of some students. We believe, however, that the difficulty index of these items might be associated with weaknesses in the students' skills about the contents of Biology, Physics and Chemistry that, not only in the ALCE, but also in TIMSS 2019 and PISA 2018, proved to be insufficient. In this sense, we recommend the use of the ALCE to assess the scientific literacy of students at the end of the 3rd cycle of Basic Education, and we encourage users to evaluate the technical quality of the instrument and to consider its suitability in order to provide valid evidence of the students' scientific literacy level in the contexts where it might be used.

**Author Contributions:** Conceptualization, M.C. (Marcelo Coppi), I.F. and M.C. (Marília Cid); methodology, M.C. (Marcelo Coppi); software, M.C. (Marcelo Coppi); validation, M.C. (Marcelo Coppi), I.F. and M.C. (Marília Cid); formal analysis, M.C. (Marcelo Coppi); investigation, M.C. (Marcelo Coppi); resources, M.C. (Marcelo Coppi), I.F. and M.C. (Marília Cid); data curation, M.C. (Marcelo Coppi); writing—original draft preparation, M.C. (Marcelo Coppi); writing—review and editing, M.C. (Marcelo Coppi); visualization, M.C. (Marcelo Coppi), I.F. and M.C. (Marília Cid); supervision, I.F. and M.C. (Marília Cid); project administration, M.C. (Marcelo Coppi); funding acquisition, M.C. (Marcelo Coppi), I.F. and M.C. (Marília Cid). All authors have read and agreed to the published version of the manuscript.

**Funding:** This work is funded by National Funds through the FCT—Foundation for Science and Technology, I.P., within the scope of the Research Grant with reference UI/BD/151034/2021 and the project UIDB/04312/2020.

**Institutional Review Board Statement:** The study was conducted in accordance with the Declaration of Helsinki, and approved by Directorate General of Innovation and Curriculum Development (protocol code 0740900001 and date of approval is 01/09/2020).

**Informed Consent Statement:** Informed consent has been obtained from the subject(s) to publish this paper.

**Data Availability Statement:** The datasets used and/or analysed during the current study are available from the corresponding author on reasonable request.

**Conflicts of Interest:** The authors declare no conflict of interest. The funders had no role in the design of the study; in the collection, analyses, or interpretation of data; in the writing of the manuscript; or in the decision to publish the results.

## Abbreviations

| | |
|---|---|
| ALCE | Avaliação da Literacia Científica Essencial |
| CS | Content of Science |
| ISTS | Impact of science and technology on society |
| NOS | Nature of science |
| PASEO | Profile of Students Leaving Compulsory Schooling |
| PISA | Programme for International Student Assessment |
| TIMSS | Trends in International Mathematics and Science Study |

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
