# Peer review of "Developing a Scientific Literacy Assessment Instrument for Portuguese 3rd Cycle Students"

_education, doi:10.3390/educsci13090941_

Round 1

Reviewer 1 Report

This article describes the development of the design of the Avaliação da Literacia Científica Essencial (ALCE) instrument, which assesses students' scientific literacy skills at the end of the 3rd cycle of basic education (Biology, Geology, Physics, and Chemistry) in cognitive areas of understanding, as well as the methodology for the process of collecting evidence of its validity. It is shown that ALCE can be a useful tool for identifying possible mismatches between learning objectives and students' science literacy levels, as well as for reviewing methodologies, lesson plans, and classroom strategies in order to modify them to better develop students' science literacy. The material has a clear structure and logical organization. For the development of ALCE, several methods were used to determine the credibility of the tool based on its content and internal structure. Methods for assessing the validity of the developed tool are scientifically substantiated. Expert feedback was used to ensure that the tool collected reliable and relevant information about students' scientific literacy levels. The list of cited references contains 57 items, but only 16 of them (28%) are publications from the last 5 years.

The topic of the article is very interesting, I congratulate the authors on this and have only a few comments.

1) Keyword search helps researchers find articles relevant to their interests and also increases readership. Thus, the heading words "Scientific Literacy" should not be used as keywords, "Learning Objectives" can be added here.

2) Review the cited literature, perhaps some elements are redundant. More recent sources can also be found (e.g. Kumar, D., Jaipurkar, R., Shekhar, A., Sikri, G., & Srinivas, V. (2021). Item analysis of multiple choice questions: A quality assurance test for an assessment tool. Medical Journal Armed Forces India, 77, S85-S89)

Reviewer 2 Report

Dear Authors,

Here are some comments:

The article titled "Developing a Scientific Literacy Assessment Instrument for Portuguese 3rd Cycle Students" presents an important endeavor in the realm of education assessment. The authors aim to develop an assessment tool for evaluating scientific literacy among 3rd cycle students in Portugal. While the study showcases valuable insights, there are certain aspects that require refinement to enhance clarity and scholarly depth. In this review, I offer constructive feedback on clarifying the research question, restructuring the presentation of research findings, and suggesting additional references related to the Nature of Science (NOS).

Clarification of Research Question:

The article would benefit from a more explicit formulation of the research question. Clearly articulating the main objectives of developing the assessment instrument and the specific aspects of scientific literacy being targeted would provide readers with a stronger understanding of the study's focus and purpose.

Restructuring the Presentation of Research Findings:

The structure of the article's presentation of research findings could be improved for better coherence and comprehension. Consider organizing the results section into distinct subsections, such as Data Analysis, Description of Results, and Evaluation. This would facilitate a clearer flow of information, making it easier for readers to follow the process of assessment tool development and its outcomes.

Inclusion of Nature of Science (NOS) References:

To enhance the theoretical underpinning of the study, the authors should incorporate additional references related to the Nature of Science (NOS). Incorporating literature that discusses NOS concepts, such as the historical development of science, the tentative and evolving nature of scientific knowledge, and the influence of cultural and societal factors on scientific inquiry, would enrich the discussion of the assessment tool's alignment with NOS principles.

In conclusion, "Developing a Scientific Literacy Assessment Instrument for Portuguese 3rd Cycle Students" holds promise as a contribution to educational assessment practices. However, to optimize its impact, the article should provide greater clarity in defining the research question, restructure the presentation of research findings for improved coherence, and bolster its theoretical foundation by including additional references on the Nature of Science. Addressing these aspects will enhance the scholarly value and readability of the article.

Best regards

Round 2

Reviewer 2 Report

The concept of "nature of science" (NOS) is very familiar to readers. If authors used Nature of Science (NS) with the same meaning, please use NOS instead of NS. If not, please clarify the difference.

Author Response

Dear reviewer, 

We agree that the acronym "NOS" is very familiar to readers and therefore we have changed all "NS" to "NOS".